# Cutaneous Squamous Cell Carcinoma: An Updated Review

**DOI:** 10.3390/cancers16101800

**Published:** 2024-05-08

**Authors:** Rina Jiang, Mike Fritz, Syril Keena T. Que

**Affiliations:** Department of Dermatology, Indiana University School of Medicine, Indianapolis, IN 46202, USA; fritzmi@iu.edu

**Keywords:** cutaneous squamous cell carcinoma, American Joint Commission on Cancer, Brigham and Women’s Hospital staging system, immunotherapy, radiation therapy, chemoprophylaxis

## Abstract

**Simple Summary:**

Incidence rates of cutaneous squamous cell carcinoma (cSCC) are projected to increase due to rising exposures to risk factors. While surgical removal continues to be the mainstay of treatment for low-risk cSCC, management of high-risk cases remains complex and lacks uniformity. This article serves as an up-to-date review of cSCC, especially highlighting high-risk patients. Topics reviewed include pathogenesis, molecular markers, and histologic subtypes, with a particular emphasis on diagnosis and management.

**Abstract:**

Representing the second most common skin cancer, the incidence and disease burden of cutaneous squamous cell carcinoma (cSCC) continues to increase. Surgical excision of the primary site effectively cures the majority of cSCC cases. However, an aggressive subset of cSCC persists with clinicopathological features that are indicative of higher recurrence, metastasis, and mortality risks. Acceleration of these features is driven by a combination of genetic and environmental factors. The past several years have seen remarkable progress in shaping the treatment landscape for advanced cSCC. Risk stratification and clinical management is a top priority. This review provides an overview of the current perspectives on cSCC with a focus on staging, treatment, and maintenance strategies, along with future research directions.

## 1. Introduction

### 1.1. Epidemiology

The estimated annual incidence of cutaneous squamous cell carcinoma surpasses one million in the United States [1]. A notable 1:1 ratio between cSCC and basal cell carcinoma (BCC) has been observed, underscoring this problem [2]. The increase has been attributed to known risk factors such as cumulative ultraviolet radiation (UV) exposure, an increasingly older population, higher rates of systemic immunosuppression, along with expanding skin cancer screenings.

However, the lack of a standardized reporting system poses a problem. Incidence and mortality rates on cSCC are not captured by U.S. national cancer registries, posing a challenge in determining the precise epidemiology in our country. For this reason, a large proportion of the literature related to this topic originates from Europe. A recent study in Germany retrieved incidence data from cancer registries of three European countries with the aim to predict trends up to 2044. From 2023 to 2024, model prediction revealed an annual percent increase ranging from 2.4% to 5.7% [3].

Aggressive cSCC is associated with a notably high risk of mortality. Mortality rates for cSCC have reached as high as that of melanoma in the southern and central regions of the United States [1]. Estimates for the United States were proposed based on available data in 2012. Among the estimated 186,157 to 419,543 white individuals who received a diagnosis of cSCC, approximately 3% of individuals with cSCC developed nodal metastases [1]. Also, an estimated 2.1% of additional cSCCs would arise from those same individuals within the same year [1]. As the population ages and exogenous immunosuppression increases, the incidence of cSCC rises. While dermoscopy enhances early detection of cSCC compared to the naked eye, its utility remains non-predictive. A comprehensive understanding is necessary to guide proper work-up and management of cSCC.

### 1.2. Pathogenesis

A majority of cSCCs originate from precursor lesions such as actinic keratoses and squamous cell carcinoma in situ (SCCIS). Supported by the concept of field cancerization, the pathogenesis of cSCCs takes a distinct course as mutations accumulate over time under the stress of an array of environmental and genetic factors, such as UV radiation [4].

At a molecular level, strong evidence regarding the tumor microenvironment (TME) of advanced cSCC has been shown to have increased levels of TGF-β, IL-10 and regulatory T cells (Treg). The infiltration of Treg into the TME attenuates anti-tumor immune responses [5]. Downregulation of plasmacytoid dendritic cells (pDCs) in advanced cSCC was also reported in comparison to well-differentiated cSCC [6]. Through tumor-associated antigen presentation, pDCs have the potential to induce anti-tumor immunity [7,8]. Together, these results provide an absence of an inflammatory microenvironment to fight advanced cSCC lesions. These add important insights into immunoregulatory mechanisms in cSCC.

Sequencing of cSCCs has noted persistent clonal expansion in NOTCH1/2, CDKN2A, HRAS, TP53, and TGF-β-R1 mutations, further emphasizing their role as drivers of tumorigenesis at initial stages [9]. Additionally, mutations in the COL11A1 gene facilitate collagen protein production with a dominant-negative effect, ultimately interrupting the architecture of the extracellular matrix [10]. This further accelerates malignant invasion via the epidermal basement membrane [10]. The mutational burden of cSCC surpasses that in lung cancer and melanoma [11,12]. Prior research has identified TP53 as the most altered tumor suppressor gene in individuals with cSCC [13]. Mutations in TP53 are involved in many human malignancies, primarily through resistance to apoptosis. Other known frequently mutated genes in cSCC are CDKN2A, Ras, and NOTCH homolog 1 [14,15]. TP53 and Ras mutations have been observed in actinic keratosis, lesions that result from cumulative sun exposure with the potential of cSCC progression [16,17,18,19].

Furthermore, tumor cell plasticity drives the shift from epithelial to mesenchymal states which significantly contributes to multidrug resistance and metastatic potential. Deletion of cadherin-related FAT1 protein in both murine and human skin cancer models have shown to promote an epithelial–mesenchymal transition (EMT) phenotype [20]. Development of such phenotype enriches tumor stemness features to enhance tumor survival.

The multifaceted pathogenesis of cSCC presents as a prominent barrier in treatment development as singular drug targets fail to cover the extensive mutational landscape.

### 1.3. Major Risk Factors

#### 1.3.1. Heritable Conditions

Several genetic factors have been linked to cSCC progression. Aside from a family history of skin cancer and blue-eyed phenotype, inherited disorders have also exhibited a heightened risk of developing cSCC due to its clear association with photosensitivity. These single-gene Mendelian disorders include xeroderma pigmentosum (XP), oculocutaneous albinism, and Kindler syndrome [21,22].

#### 1.3.2. Immune Status

Acquired immunosuppression, often a result of hematologic malignancies and prolonged immunosuppressive regimens, predisposes to cSCC. For instance, human immunodeficiency virus (HIV) infection has a 2.6-fold increase of cSCC compared to HIV-negative individuals [23]. Additionally, chronic lymphocytic leukemia (CLL) is one of the most common hematologic malignancies associated with cSCC. A 2021 systematic review demonstrated an 11.5% cSCC-associated mortality rate in CLL patients [24].

The most common skin cancer in solid organ transplant recipients (SOTR) is cSCC, with a documented 65 to 100-fold increase in incidence in comparison to the general population [25]. A 2021 systematic review found that Latin American solid organ transplant recipients have a higher prevalence of cSCC [26]. Importantly, immunosuppressed patients with metastatic cSCC have 5-year survival rates of 50–83% [27]. Profiling of cSCC tumors in immunosuppressed patients demonstrated an absence of B cells in the peritumoral stroma, a distinct feature observed in immunocompetent individuals [28].

#### 1.3.3. UV Exposure

Cumulative exposure from sunlight and tanning beds is a major factor for cSCC development, predominately in Caucasians, men, and the elderly [29]. A 2020 study found men were more likely to have cSCC on the head and neck, while women were more likely to have them on the lower limbs [30]. Despite being less common in black patients, cSCC has higher mortality in this population, at a rate of 18.4% [31]. This is much higher than the general population (4%) [32]. Delayed diagnosis of cSCC within black patients is a major contributor.

#### 1.3.4. Chemical Pollutants

Environmental exposures, such as arsenic, radon, and polycyclic aromatic hydrocarbons, were all associated with an elevated risk of cSCC development. Arsenic has traditionally been incorporated into pesticides and is occasionally found in well water. A 2010 study observed an increased level of arsenic in Asian herbal preparations [33]. Additionally, a prospective cohort study concluded that certain occupations, such as military personnel, postal workers, and public safety workers, exhibited an increased risk of developing cSCC [34].

#### 1.3.5. HPV

There is a clear association between cSCC and human papillomavirus (HPV). Oncogenes such as E6 and E7 are featured in HPV types 16 and 18, respectively, where constitutive activation drives tumorigenesis [35]. HPV 16 stands as the predominant viral subtype in ungual cSCC, highlighted by its detection in 74% of cases [36]. A 2016 meta-analysis identified additional HPV subtypes 5, 8, 17, 20, and 38 were associated with an increased risk of developing cSCC in immunocompetent individuals [37]. SOTRs often carry HPV types 8, 9, and 15 [38]. Of note, HPV is not actively transcribed in cSCC but serves as a catalyst in the initial stages of tumorigenesis.

#### 1.3.6. Drugs

Medications that can increase the risk of cSCC range from immunosuppressive agents and antimetabolites (mycophenolate mofetil, azathioprine, cyclosporine A, cyclophosphamide) to diuretics (hydrochlorothiazide) [39]. A 2018 meta-analysis revealed a dose-response risk of cSCC associated with the use of voriconazole among lung or hematopoietic transplant patients [40]. A 2022 review showed patients with myelofibrosis or polycythemia taking ruxolitinib, a Janus-Kinase (JAK) inhibitor, had an increased risk of cSCC, with the earliest case being 11 months after therapy initiation [41].

Vismodegib, a hedgehog pathway inhibitor indicated for advanced or metastatic basal cell carcinoma (BCC), was previously shown to have an 8-fold increase of cSCC, but this conclusion has not been reproducible in recent studies [42,43]. Thus, more research is warranted. The use of a v-Raf murine sarcoma viral oncogene homolog β1 (BRAF) inhibitor has revolutionized the treatment paradigm for metastatic melanoma. However, the development of cSCC in 19–26% of patients has been reported [44]. Latest research advancements have explored the utilization of MAPK (mitogen-activated protein kinase) kinase inhibitor (MEKi) with BRAF to minimize this risk factor as inhibition of upstream activators that phosphorylate ERK-1/ERK-2 in the MAPK pathway have been shown to hinder the promotion of human cSCC cell lines [45].

## 2. Recurrence and Metastatic Risk

Staging serves to risk stratify by identifying high-risk cases that may benefit from adjuvant therapy or closer monitoring. The American Joint Committee on Cancer Staging Manual (AJCC-8) and Brigham and Women’s Hospital (BWH) are the most used staging systems for cSCC (Table 1).

### Defining High Risk

Several characteristics have been connected to increased disease recurrence and poor prognosis. A 2023 meta-analysis highlighted that the highest risk for local recurrence and disease-specific death in cSCC was tumor invasion beyond the subcutaneous fat. Tumor diameter greater than or equal to 2.0 cm increases the risk of local recurrence, metastasis, and disease-specific mortality [48,49,50]. The greatest risk for distant metastases was perineural invasion [51]. A 2017 systematic review revealed that patients with clinical or radiologic evidence of perineural invasion experienced worse 5-year recurrence-free survival or disease-specific survival compared to those where perineural invasion was incidentally discovered on biopsy [52]. This suggests that when the disease manifests symptoms or appears on imaging, it may act more aggressively than if neural involvement were identified solely microscopically.

Tumor-associated factors help stratify risk and prognosis, but the definition of a high-risk cSCC lacks universal consensus. Both staging systems classify T1 as low risk. BWH T2a also appears to be low risk. BWH classifications of T2b and T3 offer a risk of nodal metastases that surpasses 20%, but metastatic risk evaluation in the AJCC-8 system remains limited [53].

Patient comorbidities, immune status, and past surgical history must be accounted for when defining high risk. A 2023 systematic review and meta-analysis of cSCC patients demonstrated immunosuppression increased the risk of local recurrence, nodal metastasis, locoregional recurrence, and all-cause mortality [51]. Another 2019 systematic review confirmed increased metastatic risk in immunosuppressed populations [54]. However, it seems the type of immunosuppression matters. HIV-positive individuals have an increased risk for locoregional recurrence, while SOTRs face a higher risk of locoregional recurrence and nodal metastases [51]. Other factors that predispose to a poor prognosis include a personal history of prior cSCC, recurrent cSCC, and cSCC arising from chronic ulcers or scars at a prior surgical site [48,50].

The histologic subtypes of cSCC differ in metastatic potential. Keratoacanthoma and verrucous carcinoma are well-differentiated subtypes of low metastatic potential. Conversely, poorly differentiated subtypes such as desmoplastic cSCC have a poor prognosis. This is supported by its highly infiltrative nature, as it has been shown to metastasize six times more often than other variants [55]. The adenosquamous variant has a more aggressive profile with high rates of recurrence and metastasis [56].

At a molecular level, studies comparing higher-risk metastatic cSCC with lower-risk cSCC or AK have found higher CpG methylation levels in metastatic cSCC compared to its precursors. Thus, incorporating methylation status in cSCC workup may predict overall survival [57]. However, these are non-standard methods. Additionally, a genetic expression test evaluating 40 genes has been validated in stratifying the risk of cSCC metastasis. Class 1 has an 8.9% risk of metastasis, class 2a has a 20.4% risk, and class 2b has a 60% risk of metastasis. However, this test was studied only in a retrospective fashion, and thus, further prospective tests are needed [58].

## 3. Approaches to Treatment

Management of most SCCIS cases involves electrodesiccation and curettage (ED&C), as supported by its high success rate, with up to a 97% cure rate depending on the body region [59,60]. For invasive lesions, surgery, with either Mohs micrographic surgery (MMS) or wide local excision (WLE), is the treatment of choice [61,62]. The risk of recurrence and metastasis have seen a significant reduction with MMS, specifically three times lower compared to standard excision [63]. For cases associated with high-risk features, MMS represents the gold standard treatment [64]. However, not all tumors are easily resectable. Important subsets ranging from large, bulky tumors to lesions that reside near critical anatomical structures intensifies case complexity. Thus, multidisciplinary approaches such as systemic therapy are vital to cSCC management.

Although there is no universal consensus to support the use of adjuvant therapy, some patients may receive adjuvant therapy in addition to surgery, particularly in high-risk cSCC. This is defined as stage T2b/T3 in the BWH staging system or T3 in the AJCC-8 system. Other considerations not in the staging systems that predispose to a higher risk include immunosuppression, lymphovascular invasion, cSCC associated with a scar or chronic inflammatory disease, or history of recurrence [65].

The benefits of adjuvant radiation or chemotherapy for cases of distinct surgical margins remain unestablished. A 2019 retrospective study of cSCC of the head and neck revealed that adjuvant radiation therapy after surgical clearance with clear margins was associated with improved overall survival (HR: 0.59; 95% CI, 0.38 to 0.90). It also reported improved disease-free survival in tumors with peripheral nerve involvement (HR: 0.47, 95% CI, 0.23 to 0.93) [66]. A 2022 retrospective study of 508 patients evaluated high T-stage cSCC treated with adjuvant radiation therapy after surgery with clear margins. The results suggest a 5-year lower cumulative incidence of local recurrence and locoregional recurrence (3.6%, 7.5%) compared to surgery with clear margins alone (8.7% and 15.3%, respectively) [67]. Furthermore, adjuvant PD-1 therapy is currently investigated in phase 3 studies such as C-POST (NCT0396004) and KEYNOTE-630 (NCT03833167) [68]. However, there have been other studies that rival these observations, showing no benefit of adjuvant radiation therapy over surgical therapy alone with clear margins [69]. Thus, the integration of adjuvant therapies has not yet become standard but is an area of ongoing research. Multidisciplinary approaches to complex cases remain crucial and should be highly considered.

### 3.1. Immune Checkpoint (Anti-PD1) Inhibitors

Prior to the emergence of EGFR and PD-1 inhibitors, cytotoxic chemotherapeutic agents such as doxorubicin, cisplatin, and paclitaxel have yielded high efficacy in treating cSCC [70]. However, concerns surrounding temporary response rates due to acquired resistance and treatment-related toxicities have raised concerns.

There are two FDA-approved PD-1 inhibitors utilized for the therapy of cSCC: cemiplimab and pembrolizumab [71,72]. Response to PD-1 inhibition ranges from 34 to 52% for unresectable stage la disease and metastatic disease [73]. In the EMPOWER studies, Cemiplimab yielded objective responses in 44% of the sample population [71,72]. In addition, the CARKSIN trial found that, for patients with PD-L1+ staining cSCC tumors, there was an objective response rate (ORR) of 55% to pembrolizumab, while PD-L1- staining tumors had an ORR of only 17% to pembrolizumab [74]. This further emphasizes the notion of considering staining for PD-L1+ during tumor work-up [51]. Tumors with a high mutational burden have shown to be more responsive to PD-1 inhibitors compared to those that have a lower mutational burden. Currently, PD-1 inhibitor treatment guidelines for patients with cSCC do not account for mutational burden [71,72,75,76,77,78].

A 2020 systematic review of 131 patients treated with PD-1 inhibitors for locally advanced, regionally metastatic, and distant metastatic cSCC presented a complete response in 10% of cases and a partial response in 50% of cases. Notably, 60% and 79% had radiation and chemotherapy, respectively, prior to PD-1 inhibition [79]. Interestingly, a large study of melanoma patients treated with PD-1 inhibitors showed that the risk of progression decreased by 13% for each decade of age [80].

Limited data exists for PD-1 inhibitor use in transplant patients. A 2020 systematic review evaluated 57 transplant patients on immune checkpoint inhibitors for advanced malignancies; 37% experienced organ rejection, and 14% died from rejection [81]. Kidney patients were most affected, followed by the liver then the heart. The overall response rate for PD-1 inhibitors was 40% [81]. This study demonstrated the imbalance of risk versus benefit when considering checkpoint inhibitors in transplant patients. HIV patients showed a response to PD-1 therapy without unexpected adverse events. It had no effects on HIV viremia or CD4+ count [80,82]. Stratifying patients based on immunosuppression type is crucial when considering their risk and treatment options.

Overall, the PD-1 inhibitors are well-tolerated, but grade 3 toxic effects have been reported in 6% to 51% of patients [71,72,75,76,77,78]. Adverse events observed in any grade include diarrhea (27%), fatigue (27%), nausea (17%), rash (15%), and constipation (15%), with many side effects linked to autoimmune reactions [71].

### 3.2. EGFR Inhibitors

The high surface expression of epidermal growth factor receptors (EGFR) on cSCCs prompted the development of targeted therapy against this Ras-Raf-mitogen-activated protein kinase pathway, with cetuximab being a therapeutic option for advanced cSCCs [83,84].

A 2023 systematic review of EGFR inhibitors in advanced cSCC revealed a modest response rate of 28%, with a mean progression-free survival of 4.8 months and an overall survival of 11.7 months [84,85]. This is notably inferior to Programmed cell death protein-1 (PD-1) inhibitor therapy. However, it may be useful in patients who are not candidates for PD-1 inhibitors, including SOTRs or those with uncontrolled autoimmune disorders. When combined with radiation or cisplatin therapy, cetuximab, an EGFR inhibitor, had a 50–78% response rate. Unfortunately, the response was not sustained for more than 14.6 months [86,87,88]. The I-TACKLE trial studied EGFR inhibitors combined with PD-1 inhibitors: cetuximab plus pembrolizumab. This had an overall response rate of 38%, which was superior to pembrolizumab alone (34%) in locally advanced/metastatic cSCC [89]. Of note, combining EGFR and PD-1 inhibitors in other squamous cell cancers (i.e., non-small cell lung cancer) has shown high-grade adverse effects such as hepatotoxicity [90,91].

### 3.3. Antiviral Therapies (HPV Vaccines)

While alpha-HPV is responsible for most cSCCs, recent evidence has shown beta-HPV to contribute to cSCC tumorigenesis as well. Current investigations on beta-HPV vaccines are underway to aid in targeted therapy of newly discovered strains [92].

### 3.4. Other Emerging Therapies

There are currently no drugs designed for the treatment of cSCC specifically. However, histone deacetylase inhibitors (vorinostat, remetinostat, abexinostat) are currently being investigated [93,94]. Acetylation of histones via histone deacetylase (HDAC) enzymes plays a key role in cSCC-related gene regulation [93,94]. Dysregulation of HDACs drives tumorigenesis [95]. Thus, inhibition of HDACs has been shown to induce apoptosis in tumor cells through the accumulation of reactive oxygen species (ROS) [95].

Investigations regarding combination therapies are underway. This includes pairings between EGFR inhibitors, Anti-PD1 inhibitors, HDAC inhibitors (abexinostat), and radiation therapy (NCT03944941) (NCT03590054) (NCT03666325). Additional clinical trials on immune checkpoint blockade are currently under investigation as well. Further trials on immune checkpoint blockade include anti-CTLA4 inhibitors such as ipilimumab and tremelimumab as neoadjuvants in treating cSCC. (NCT04620200) (NCT03450967) [96].

Talimogene laherparepvec (TVEC) is an oncolytic immunotherapy formulated through genetic modification of herpes simplex virus type 1 (HSV-1) [97]. The mechanism of action aims to selectively replicate the virus and propagate it in tumor cells to increase antigen presentation on MHC-I cells, allowing for more tumor-antigen presentation by dendritic cells [97]. TVEC was shown to temporize the advancement of a cSCC in a SOTR in one case report, and ongoing investigations through Phase Ib/Phase II trials are currently underway to further examine its utility [98,99,100]. Table 2 provides a comprehensive summary of the more recent therapies for cSCC as discussed above.

PI3K/mTOR inhibitors have demonstrated safety in patients with squamous cell carcinoma in situ (SCCIS), although efficacy endpoints were not met (NCT03333694).

Other non-medical or surgical approaches have demonstrated efficacy in treating microinvasive and invasive SCC, including ablative fractional lasers and plum blossom needle treatments [101]. In a phase II study involving 25 patients, injection of the photosensitizer temoporfin achieved a complete response in 96% of cases [102]. Future possibilities include the utilization of nanoparticles to facilitate deeper penetration of photosensitizers, targeting tumors, and enhancing the overall efficacy of PDT [103].

**Table 2 cancers-16-01800-t002:** Newer and emerging therapies for cSCC.

Therapeutic Agent	Study Population	Mechanism of Action	Adverse Effects	Clinical Trial
EGFR Inhibitor				
Cetuximab [84,104]	Locally advanced or recurrent/metastatic cSCC	Chimeric monoclonal antibody against EGFR	Infusion related reactions, acneiform rash, pruritis, infection, GI discomfort	NCT03325738
Panitumumab, [105]	Locally advanced or recurrent/metastatic cSCC	Humanized monoclonal antibody against EGFR	Fatigue, acneiform rash	
Lapatinib, [106]	Neoadjuvant therapy for locally advanced or recurrent/metastatic cSCC	Small-molecule TKI	Diarrhea, rash, pancreatitis	NCT0166431
Erlotinib, [107]	Nonresectable locally advanced or recurrent/metastatic cSCC	Small-molecule TKI	Acneiform rash, diarrhea	NCT01198028
Gefitinib, [108]	Locally advanced or recurrent/metastatic cSCC	Small-molecule TKI	Acneiform rash, diarrhea, fatigue, nausea	
Immunotherapy				
Cemiplimab, [75]	Locally advanced or recurrent/metastatic cSCC in unresectable setting	Anti-PD1 Inhibitor: Engineered humanized IgG4 monoclonal antibody that binds to PD-1 and blocks ligand interaction between PD-L1 & PD-L2	Fatigue, pruritus, diarrhea, hypothyroidism, arthralgia	EMPOWER-CSCC
Pembrolizumab, [74,76]	Locally advanced or recurrent/metastatic cSCC in unresectable setting	Anti-PD1 Inhibitor: Engineered humanized IgG4 monoclonal antibody that binds to PD-1 and blocks ligand interaction between PD-L1 & PD-L2	Fatigue, pruritus, diarrhea, asthenia, hypothyroidism, pneumonitis	KEYNOTE-629 CARSKIN
Talimogene laherparepvec (TVEC) **^†^,** [97]	Recurrent cSCC following SOTR (case report), Low risk cSCC	Genetically modified herpes simplex virus 1 that selectively replicates in tumor cells to promote tumor-antigen presentation	Thrombocytopenia, transient fatigue, flu-like symptoms, headache	NCT04349436
HDAC Inhibitors, [93,94,95]				
Vorinostat **^†^**	Concurrent radiation therapy in Stage III, IVa, IVb HNSCC	Inhibits histone deacetylation to repress gene transcription	Anemia, leukopenia, weight loss, mucositis, xerostomia, nausea, hyponatremia, dysphagia	NCT01064921
Remetinostat **^†^** Abexinostat **^†,‡^**	Neoadjuvant for non-invasive cSCC Combined with Pembrolizumab for advanced solid tumor malignancies	Inhibits histone deacetylation to repress gene transcription Inhibits histone deacetylation to repress gene transcription	No reported systemic adverse effects N/A	NCT03875859 NCT03590054
Anti-CTLA4 Inhibitors				
Ipilimumab **^†^,** [109,110]	Neoadjuvant in advanced cSCC prior to surgery, allograft patients	Antibody against CTLA-4 to downregulate T-cell activation and proinflammatory cytokine release	Morbilliform rash, pruritus	NCT04620200 NCT03816332
Tremelimumab **^†^,** [96]	Neoadjuvant for recurrent or metastatic HNSCC	Antibody against CTLA-4 to downregulate T-cell activation and proinflammatory cytokine release	Anemia, constipation, pneumonia, electrolyte imbalances, hyperglycemia	NCT03450967
PI3K/mTOR Inhibitors				
CLL442 *^,^**^†^**	SCCis	Inhibits PI3K/mTOR pathway to downregulate cell migration and lymphocyte differentiation	No severe adverse effects reported with topical application	NCT03333694

Abbreviations: EGFR, epidermal growth factor receptor; TKI, tyrosine kinase inhibitor; PD1, programmed death-1; PD-L1/2, programmed death ligand 1/2; cSCC, cutaneous squamous cell carcinoma; SOTR, solid organ transplant recipient; HDAC, histone deacetylase; CTLA4, cytotoxic T-lymphocyte-associated antigen 4; HNSCC, head & neck squamous cell carcinoma; PI3K/mTOR, phosphatidylinositol-3-kinase/mammalian target of rapamycin; SCCis, squamous cell carcinoma in situ; * Primary outcome of lesion clearance not achieved; ^†^ Emerging therapeutic strategies in early-stage clinical trials; ^‡^ Ongoing clinical trial with no reported data yet.

## 4. Preventative Measures and Chemoprophylaxis

For the general population, sun protection measures, including sunscreen, clothing barriers, and sun avoidance at peak hours, are recommended for cSCC prevention, along with follow-up dermatologic exams. High-risk populations consisting of SOTRs, individuals with hematological malignancies, and other immunosuppressed populations may require secondary chemoprevention measures. Specifically, chemoprophylaxis may be warranted for patients who have had ≥ five non-SSCIS or one high-stage cSCC. Various agents, such as 5-fluorouracil, resveratrol, vitamin D, and methotrexate, have shown improved effectiveness against SCC cells in both in vitro and in vivo settings [111,112,113,114].

Two widely used chemoprophylaxis methods involve topical 5-fluorouracil (5-FU) and photodynamic therapy (PDT), specifically for precancerous lesions such as AKs. A 2018 randomized trial involving 932 veterans indicated a 75% drastic risk reduction in cSCC development for 1 year following the application of 5-FU topical cream twice daily for 2–4 weeks on the face and ears [115].

PDT utilizes topical photosensitizers such as 5-aminolevulinic acid (5-ALA) combined with phototherapy to produce ROS with the aim of eliminating the proliferating cells driving AKs and SCCIS [116]. It is FDA-approved to treat AKs and other superficial dermatologic neoplasms. Close monitoring is needed for this modality, as sun exposure can exacerbate adverse effects of blistering, erythema, and pain [117,118,119]. When comparing the effectiveness between topical 5-FU and PDT in patients with SCCIS, reported response rates at the 12-month mark highlighted a higher value for PDT (82%) over topical 5-FU (48%) [120].

Oral therapeutic options include retinoids, such as acitretin or isotretinoin. Retinoids contribute greatly to cSCC chemoprophylaxis through multiple mechanisms pertaining to immunomodulation and cell cycle control [103,121]. Although all mechanisms are not fully understood, several studies have highlighted a reduction in cSCC development in particular immunosuppressed cohorts such as patients with XP and SOTRs [122,123,124,125]. Treatment is initiated based on the following criteria: > five cSCCs within 2–3 years, development of UV-related T2b/T3 cSCC, and cSCCs that are unresponsive to 5-FU or PDT [65]. Close monitoring for adverse effects is crucial. This includes ordering a baseline complete blood cell count, serum creatinine, lipid panel, and liver function tests (LFTs). Monthly dosage increases require lipid panels and LFTs. Once the patient is on a stable dose, lipids and LFTs should be conducted every three months [65]. Patients with renal dysfunction or a kidney transplant should have creatinine monitored [126]. Treatment adherence remains imperative as the therapeutic effects of acitretin diminish rapidly following discontinuation [127].

Nicotinamide, a form of vitamin B3, is another oral agent for chemoprevention of cSCC. It is thought to repair UV-associated DNA damage through a series of redox reactions [128,129]. In one study in an immunocompetent population, it reduced cSCC burden by 30% [130]. However, a study of SOTRs taking nicotinamide showed no statistical difference in reducing cSCCs, but this study was notably limited by an underpowered design [131]. Despite inconsistent results, its low cost and safety profile in comparison to acitretin is appealing [132]. While the trials listed previously reported no adverse effects, caution is advised as high doses (> 3 g/day) can result in reversible hepatotoxicity [133]. Given the premature conclusions drawn from limited studies, larger trials are needed to evaluate the benefits of nicotinamide chemoprophylaxis in SOTRs.

Multi-disciplinary management is essential in chemopreventive measures for SOTRs to mitigate risks. When there is a heightened concern for cSCC, a major consideration is to replace immunosuppressants with sirolimus. Sirolimus is an inhibitor of the mammalian target of rapamycin (mTOR), a central component in stimulating mitogenic cell division [134]. Studies have indicated a significantly lower susceptibility to cSCC among SOTRs with sirolimus as a part of their regimen [135,136]. Sirolimus side effects most frequently include hyperlipidemia and myelosuppression. Other side effects include peripheral edema, rash, and abdominal pain [137,138]. It should be noted that fewer events have been reported when following gradual conversion protocols [137,138].

Capecitabine is another immunosuppressant option in patients with multiple cSCCs. Capecitabine is a prodrug of 5-FU that inhibits thymidylate synthase, further disrupting RNA and DNA biosynthesis [139]. As a prodrug, the safety profile varies slightly from that of 5-FU, with more hand–foot syndrome than gastrointestinal discomfort [140]. The benefits of capecitabine are clear, with utility in preventing AKs and SCCIS. It is imperative that individuals are screened for dihydropyrimidine dehydrogenase deficiency, as the use of capecitabine may lead to systemic toxicity and death [141]. Previous studies have indicated potential in prevention efficacy with a 68% reduction of cSCC per month over one year in SOTRs [142].

Lastly, SRC family kinase (SFK) inhibitors have shown promising results as a method of primary prevention. A highly investigated oncogenic cascade, phosphatidylinositol 3-kinase (PI3K) is one of the downstream kinases of the EGFR pathway. Overactivation of the PI3K signaling pathway has been observed in cSCC [143]. As a lipid kinase, its involvement in the cell cycle and survival is based on its role in modulating downstream signaling through the recruitment of cytoplasmic proteins to the membrane surface [144]. Early-phase clinical trials focused on targeting SRC family kinases have proven promising in treating AKs, ultimately preventing the transition to cSCC. Tirbanibulin is approved for the treatment of AKs. It inhibits SRC family kinases, with downstream activity against the PI3K pathway. A 2023 study showed complete clearance of AKs in patients treated with tirbanibulin was achieved in 47% of cases upon first follow-up and 57% during the second visit [139]. Adverse effects predominantly involved local reactions, with erythema (80%) and flaking/scaling (43%) that resolved spontaneously [145]. The favorable safety profile of tirbanibulin highlights a novel application in minimizing the risk of malignant transformation of AK lesions to cSCCs.

## 5. Conclusions

With an aging population and increasing exposures to risk factors, cSCC incidence rates are predicted to rise globally. While surgical destruction stands as the first line of treatment for low-risk cSCC, the management of high-risk cSCC remains complex and non-uniform. The two most common staging criteria include the AJCC-8 and BWH systems. Driven by the enhanced understanding of molecular and genetic perspectives, the past decade has witnessed several advances in the evolving management landscape of cSCC. EGFR inhibitors and anti-PD1 inhibitors are two new options for unresectable cSCC, with PD-1 inhibitors having far greater efficacy. Future avenues in this field include a widening of vaccines against HPV subtypes and TVEC, along with targeting PI3K/mTOR pathways for the treatment of SCCIS.

## Figures and Tables

**Table 1 cancers-16-01800-t001:** American Joint Committee on Cancer (AJCC) cutaneous SCC staging system for tumors of the head and neck skin 8th edition ^¥^.

T Category	T Criteria	N Category	N Criteria for Pathologic N	M Category	M Criteria
TX	Primary tumor cannot be identified	NX	Regional lymph nodes cannot be assessed	M0 M1	No distant metastasis Distant metastasis
Tis	Carcinoma in situ	N0	No regional lymph node metastasis		
T1	Tumor < 2 cm in greatest dimension	N1	Metastasis in a single ipsilateral lymph node, ≤3 cm in greatest dimension and ENE− *		
T2	Tumor ≥ 2 cm but <4 cm in greatest dimension	N2	Metastasis in a single ipsilateral lymph node ≤3 cm in greatest dimension and ENE+; or >3 cm but not >6 cm in greatest dimension and ENE−; or metastases in multiple ipsilateral lymph nodes, none >6 cm in greatest dimension and ENE−; or in bilateral or contralateral lymph nodes, none > 6 cm in greatest dimension and ENE−		
T3	Tumor ≥ 4 cm in clinical diameter OR minor bone erosion OR perineural invasion OR deep invasion ^†^	N2a	Metastasis in single ipsilateral or contralateral node ≤3 cm in greatest dimension and ENE+; or in a single ipsilateral node > 3 cm but not > 6 cm in greatest dimension and ENE−		
T4	Tumor with gross cortical bone/marrow, skull base invasion, and/or skull base foramen invasion	N2b	Metastasis in multiple ipsilateral nodes, none > 6 cm in greatest dimension and ENE−		
T4a	Tumor with gross cortical bone/marrow invasion	N2c	Metastasis in bilateral or contralateral lymph nodes, none > 6 cm in greatest dimension and ENE−		
T4b	Tumor with skull base invasion and/or skull base foramen involvement	N3	Metastasis in a lymph node > 6 cm in greatest dimension and ENE−; or in a single ipsilateral node >3 cm in greatest dimension and ENE+; or multiple ipsilateral, contralateral, or bilateral nodes, any with ENE+		
		N3a	Metastasis in a lymph node >6 cm in greatest dimension and ENE−		
		N3b	Metastasis in a single ipsilateral node >3 cm in greatest dimension and ENE+; or multiple ipsilateral, contralateral, or bilateral nodes, any with ENE+		
**Brigham and Women’s Hospital Tumor Staging System ^£^**					
**Stage**		**No. of High-Risk Factors ^‡^**			
T1		0			
T2a		1			
T2b		2–3			
T3		≥4			

ENE, Extranodal Extension; * Extension through the lymph node capsule into surrounding connective tissue, with or without stromal reaction. † Deep invasion is defined as invasion beyond subcutaneous fat or 6 mm (as measured from granular layer of adjacent normal epidermis to the base of the tumor). Perineural invasion is defined as tumor cells within the nerve sheath of a nerve deeper than the dermis or measuring ≥0.1 mm or presenting with clinical or radiographic involvement of named nerves without skull base invasion. ^‡^ Brigham and Women’s Hospital high-risk factors include tumor diameter ≥2 cm, poorly differentiated histology, perineural invasion ≥0.1 mm, or tumor invasion beyond the subcutaneous fat (excluding bone invasion which automatically upgrades tumor to Brigham and Women’s Hospital stage T3). ¥ Obtained with permission from AJCC Cancer Staging Manual, 8th edition, Springer International Publishing, New York, New York, © 2017 [46]. £ Obtained with permission from Que et al., 2018 [47].

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
