# Peer review of "Cutaneous Squamous Cell Carcinoma: An Updated Review"

_cancers, 2024, doi:10.3390/cancers16101800_

Round 1

Reviewer 1 Report

Comments and Suggestions for Authors

To the editor,

The authors have submitted a review article on cutaneous squamous cell carcinoma (CSCC); this is an important an often misunderstood topic, but one of high clinical importance, as it is among the 2-3 most common cancers worldwide, and also a complex tumor type that require multi-disciplinary care to achieve the best outcomes.  The authors have done an excellent job in covering the challengingly wide range of topics related to CSCC, albeit in a concise manner.  Below, we have submitted comments which we hope will improve the manuscript for the reader.

Introduction 1.1
-during the 3rd paragraph, when describing incidence of CSCC in the US and the proportion with nodal metastases.  It would also be recommended to highlight first that CSCC itself can become a disease with high morbidity and mortality.  Many clinicians often mistake CSCC as being a superficial tumor that cannot metastasize or cause death.  For example, consider commenting on mortality rates for CSCC, which some estimate may be as high as that of melanoma.

Pathogenesis 1.2
-during the 1st paragraph, would also highlight that CSCC tumors often show mutations associated with UV radiation

Immune Status 1.3.2
-there is a nice paragraph discussing solid organ transplant recipients, but rather little about HIV or hematologic malignancies as sources of immune suppression -> please expand on this.  These would be helpful for the reader.  Please specify which hematologic malignancies are associated with CSCC.
-typo: remove underscore next to title of this section

Drugs 1.3.6
-the comment on BRAF inhibition as a risk factor for CSCC is important; should be noted that use of MEK inhibition with BRAF can alleviate this; also consider a discussion of how drugging the MAPK pathway drives CSCC. 

Recurrence and Metastatic Risk 2.
Defining High Risk 3.
-The numbering/headers used here are a bit odd.  Recurrence/Metastatic Risk and High Risk seem like the same topic.  Consider renumbering to have Recurrence and Met Risk as 2., then Defining High risk as 2.1

Defining High Risk 3.
-The 4th paragraph discussing predictive biomarkers would fit better aft the following paragraph that discusses histology.
-This same 4th paragraph discusses potential future areas of interest (methylation status, gene expression profiling); the tone / language here should be clear that these are at not standard approaches.

Approaches to Treatment 4.
-the 1st paragraph discusses surgical approaches as an option for localized lesions.  However, this paragraph should also discuss that not all tumors can easily be resected, for example, large bulky tumors, lesions that involve bone, or lesions located in close proximity with vital structures, etc.  Thus it should be pointed out that multi-disciplinary care is helpful and that some tumors may not be resectable, thus requiring other approaches, i.e. systemic therapy.
-the first line of the 2nd paragraph says that “some patients need adjuvant therapy”.  Given that adjuvant therapy is still not a standardized approach as considered later, it may be better to state that there is no consensus around adjuvant therapy.  Again would also stress the need for multi-disciplinary approaches to complex cases.
-in the 3rd paragraph, would also state that adjuvant PD-1 therapy is being studied in at least 2 phase 3 studies: C-POST and KEYNOTE-630

EGFR 4.1 and PD-1 4.2
-please switch the order of these therapies.  Although EGFR inhibitors have been studied for longer, only PD-1 is FDA approved for CSCC, and is considered the standard first line systemic therapy.  EGFR should be considered if PD-1 fails, or if patients are not eligible for PD-1.

EGFR 4.1
-the second paragraph should cite J Clin Oncol 2011 Maubec, which is a prospective phase II study of EGFR therapy for CSCC.

PD-1 4.2
-in addition to the CARSKIN study, the 1st paragraph should also discuss the EMPOWER studies (N Engl J Med 2018 Migden and Lancet Oncol 2020 Migden, already cited), which are also prospective phase II studies of PD-1 for CSCC.

Other Emerging Therapies 4.4
-besides these therapies listed, cytotoxic chemotherapy is also an available and effective treatment for CSCC.  The authors should at least mention that drugs such as doxorubicin, carboplatin and paclitaxel have all been used in treating CSCC prior to development of EGFR or PD-1, and thus can also be considered.  This would not go under “emerging”, but perhaps “chemotherapy”.

Preventive Measures and Chemoprophylaxis 5.
-in the 1st paragraph, beyond sun protection, follow-up with dermatology/skin exams should at least be mentioned as important preventive measures.
-in the 4th paragraph, the discussion of laser ablation and other approaches to treat existing SCC lesions should be moved to the paragraph on treatment, and not prevention.

Reviewer 2 Report

Comments and Suggestions for Authors

An excellent review paper.

Well written and concise.

Early detection of cSCC is enhanced when clinicians acquire higher skill and use dermoscopy.  This could be mentioned.

Reflectance Confocal Microscopy is typically not as useful because the common presence of surface keratin scatters the incident light reducing image resolution.  

Reviewer 3 Report

Comments and Suggestions for Authors

This is a well-written comprehensive review on cSCC.

I only have one concern:

I propose to include a chapter on tumor cell plasticity leading to the transition of epithelial to mesenchymal and notably stem cell-like tumor cell phenotypes that pronouncedly contribute to multidrug resistance, disease recurrence and metastasis.

Author Response

Please see the attachment, thank you
